# Cytotoxic Activity, Topoisomerase I Inhibition and In Silico Studies of New Sesquiterpene-aryl Ester Derivatives of (-) Drimenol

**DOI:** 10.3390/molecules28093959

**Published:** 2023-05-08

**Authors:** Ileana Araque, Javiera Ramírez, Rut Vergara, Jaime Mella, Pablo Aránguiz, Luis Espinoza, Waleska Vera, Iván Montenegro, Cristian O. Salas, Joan Villena, Mauricio A. Cuellar

**Affiliations:** 1Facultad de Farmacia, Escuela de Química y Farmacia, Universidad de Valparaíso, Av. Gran Bretaña 1093, Valparaíso 2340000, Chile; ileana.araque@postgrado.uv.cl (I.A.); javiera.rp6@gmail.com (J.R.); waleska.vera@uv.cl (W.V.); 2Centro de Investigaciones Biomédicas, Escuela de Medicina, Facultad de Medicina, Universidad de Valparaíso, Viña del Mar 2520000, Chile; rut.vergara@postgrado.uv.cl (R.V.); juan.villena@uv.cl (J.V.); 3Instituto de Química y Bioquímica, Facultad de Ciencias, Universidad de Valparaíso, Valparaíso 2340000, Chile; jaime.mella@uv.cl; 4Centro de Investigación Farmacopea Chilena (CIFAR), Universidad de Valparaíso, Valparaíso 2340000, Chile; ivan.montenegro@uv.cl; 5Escuela de Química y Farmacia, Facultad de Medicina, Universidad Andrés Bello, Viña del Mar 2520000, Chile; pablo.aranguiz@unab.cl; 6Departamento de Química, Universidad Técnica Federico Santa María, Avenida España 1680, Valparaíso 2340000, Chile; luis.espinozac@usm.cl; 7Facultad de Medicina, Escuela de Obstetricia y Puericultura, Universidad de Valparaíso, Angamos 655, Reñaca, Viña del Mar 2520000, Chile; 8Departamento de Química Orgánica, Facultad de Química y de Farmacia, Pontificia Universidad Católica de Chile, Avenida Vicuña Mackenna 4860, Macul, Santiago de Chile 7820436, Chile; cosalas@uc.cl

**Keywords:** drimenol, aryl-sesquiterpene esters, cytotoxic activity, caspase, topoisomerase I, docking, molecular dynamic

## Abstract

In this study, we aimed to evaluate two sets of sesquiterpene-aryl derivatives linked by an ester bond, their cytotoxic activities, and their capacity to activate caspases 3/7 and inhibit human topoisomerase I (TOP1). A total of 13 compounds were synthesized from the natural sesquiterpene (-)-drimenol and their cytotoxic activity was evaluated in vitro against three cancer cell lines: PC-3 (prostate cancer), HT-29 (colon cancer), MCF-7 (breast cancer), and an immortalized non-tumoral cell line (MCF-10). From the results, it was observed that **6a** was the most promising compound due to its cytotoxic effect on three cancer cell lines and its selectivity, **6a** was 100-fold more selective than 5-FU in MCF-7 and 20-fold in PC-3. It was observed that **6a** also induced apoptosis by caspases 3/7 activity using a Capsase-Glo-3/7 assay kit and inhibited TOP1. A possible binding mode of **6a** in a complex with TOP1-DNA was proposed by docking and molecular dynamics studies. In addition, **6a** was predicted to have a good pharmacokinetic profile for oral administration. Therefore, through this study, it was demonstrated that the drimane scaffold should be considered in the search of new antitumoral agents.

## 1. Introduction

Drimenol (Figure 1I) is a natural compound with bicyclic farnesane-type skeleton that can be found in different species of *Drimys* (Winteraceae) and *Polygonum* (Polygonaceae) genus [1,2]. This metabolite shows several biological activities including cytotoxic [3], antibacterial [4], and antifungal activity [5,6]. Synthetically, it is useful as a starting compound or intermediate for the synthesis of several natural drimanes, merosesquiterpenes, and related compounds [7]. Merosesquiterpenoids are compounds of mixed biogenesis based on farnesane-type skeleton and together with an aromatic moiety with variety in oxidation degrees and changes in their substitution patterns, generally with a *para*- or *ortho*-hydroquinone systems, and these are commonly found in algae and/or marine sponges. Many marine sesquiterpenoid hydroquinones are of considerable interest for their versatile biological activities, such as antimicrobial, antiviral, immunomodulatory, and cytotoxic [8]. For example, widendiol A (Figure 1II), isolated from the marine sponge *Xestospongia wiedenmayeri*, inhibits cholesterol ester transfer protein (CETP) [9]. Other related metabolites are dactylospontriol (III) and smenodiol (IV); the first was isolated from *Albatrellus confluens* and the latter from the sponge *Smenospongia* sp. [10,11]. These compounds were tested for their cytotoxic potential in three colorectal cancer cell models HCT116, RKO, and HT29, where III demonstrated activity in HCT116, RKO, and HT29 cells and IV showed moderate activity in these studies [12].

Moreover, it was reported that terrestrial fungi can produce a similar group of compounds that differ from marine metabolites. Albaconol (V) and neoalbaconol (VI) are prenylated resorcinol isolated from fresh fruiting bodies of the basidiomycetes *Albatrellus confluens* that showed interesting biological activity [13]. Albaconol inhibited the proliferation of K562, A549, BGC-823, and Bcap-37. In addition, neoalbaconol was reported to be an inhibitor of the protein kinase PDK1 that induces different forms of cell death by affecting cellular energy metabolism through the PDK1-PI3k-Akt pathway [14].

One of the mechanisms of action proposed for the cytotoxic effect of albacanol was the inhibition of topoisomerase II (TOP2), where the results showed that this compound significantly influenced TOP2 activity in an apparently dose-dependent manner, determined by measuring ATP-dependent relaxation of supercoiled pBR322 DNA [15]. The significance of this result is that TOP2 and TOP1 are crucial enzymes that control DNA topology to facilitate chromosome segregation in mitosis, acting during DNA replication, transcription, recombination, and chromosome decatenation [16]. To perform this, TOP1 transiently cleaves a single strand of the DNA double helix, while TOP2 transiently cleaves both strands [17]. Stabilization of the covalent enzyme-DNA complexes by topoisomerase inhibitors leads to DNA damage and, thus, apoptosis induction. Because of this, TOP1/2 are pharmacological targets of drugs in clinical use, such as the TOP1 inhibitors camptothecin, irinotecan, and topotecan [18], and TOP2 inhibitors daunorubicin, doxorubicin, and etoposide [19]. Daunorubicin and doxorubicin were the first anthracyclines approved as anticancer drugs in 1967 and 1974, respectively, and since then, they were found to be among the most widely used cancer chemotherapeutic agents worldwide [20]. Despite their impressive potential, the usefulness of both compounds is hampered by several acute and long-term adverse effects. The most severe is cumulative cardiotoxicity since it limits the dosage that can be prescribed [21]. Therefore, topoisomerase inhibitors are interesting and possible targets for new therapeutic approaches against cancer in the form of chemotherapeutic drugs.

It is worth considering that, in medicinal chemistry, research efforts were directed towards obtaining new analogs through more efficient synthetic strategies that include structural variants of those already obtained from natural sources. All the bioactive sesquiterpene derivatives shown in Figure 1 provide chemical fragments to design novel anticancer agents, and all of them can be obtained from (-) drimenol. In this work, we synthesized two new series of sesquiterpen-aryl compounds (Figure 2). The design of the compounds of both series was based on some modifications of the sesquiterpene scaffold: i) substitution of the methylene linker by an ester group, which differs in the pattern that links to the arene moiety; ii) position of the double bond in the drimane skeleton; and iii) substitution pattern in the arene ring by some groups present in these natural products such as methyl, hydroxyl, methyl ester. Subsequently, the biological effect on three cancer cell lines and one non-neoplastic cell as control of all synthesized compounds were evaluated. In addition, to determine a possible mechanism of action of these sesquiterpene derivatives, their capacity to inhibit human TOP1 and apoptosis induction by caspases 3/7 activity were performed. Molecular docking and dynamic molecular studies were carried out on TOP1 to understand an eventual binding mode for the most promising compound.

## 2. Results and Discussion

### 2.1. Synthesis

To obtain the two new series of sesquiterpenoids ester analogues **6a**–**f** and **8a**–**g**, the synthetic strategy shown in Figure 1 was developed. For the synthesis of compounds **6a**–**f**, drimenic acid **3** is required. When (-) drimenol **1**, obtained from the hexane extract of the bark of *Drimys winteri* Forst [22], was treated with pyridinium chlorochromate (PCC) in dichloromethane at 0 °C. it produced aldehyde **2** in good yield (78%). Unsaturated aldehyde **2** was oxidized according to the methodology described by Aricu [23] to yield the drimenic acid **3** (85%). The reaction of **3** with thionyl chloride (SOCl_2_) and pyridine produced in situ acid chloride **4**, which reacted with phenols **5a**–**f** to give the corresponding esters **6a**–**f** with yields in the order of 28 to 70%. The isomerization of the C_7_–C_8_ double bond into C_8_–C_9_ during the esterification reaction was previously observed in sesquiterpen-aryl amides synthesis [23]. On the other hand, through simple esterification of **1** and several acid benzoic chlorides **7a**–**g**, it was possible to obtain **8a**–**g** with low to excellent yields (35 to 94%).

All synthesized compounds (intermediates and final compounds) were purified by column chromatography with their respective mobile phase, and their structures were established from their spectral properties (MS, ^1^H NMR, and ^13^C NMR, see the experimental section and Appendix A).

### 2.2. Biological Evaluation

To establish whether synthesized sesquiterpene derivatives possess potential antitumor properties, their cytotoxic effects were determined in vitro in a panel of human cancer cell lines [24] and their effects on a non-neoplastic cell line as a control. For this purpose, a colorimetric sulforhodamine B assay was performed to quantify the cytotoxic effect through their respective IC_50_ values [25,26]. Each compound’s serial dilutions (12.5, 25, 50, and 100 µM) were evaluated in triplicate. In our study, the human cancer cell lines used were MCF-7 (breast adenocarcinoma); HT-29 (colorectal adenocarcinoma); PC-3 (prostate adenocarcinoma); and MCF-10 control cells (normal breast epithelium). For comparative purposes, the literature IC_50_ values of daunorubicin and 5-fluorouracil (5-FU), measured in the same cell lines and under the same experimental conditions, were included [27,28,29]. Similarly, drimenol (**1**) was included for comparative purposes [28].

Table 1 shows the IC_50_ values of compounds **6a**–**f** and **8a**–**g** against cancer cell lines and MCF-10 cells. Overall, the cytotoxicity of the sesquiterpene-aryl derivatives was homogeneous. However, **6b**–**d** and **8a**–**b** were not able to inhibit the growth of cancer cell lines. Interestingly, the base sesquiterpene system of these derivatives, drimenol, was inactive in two cell lines, as reported by Montenegro et al. Therefore, it was shown that the incorporation of the aromatic ring in the targeted products increased cytotoxicity in most cases. Likewise, the cytotoxicity of the reference drugs depended on the line considered. Since a potential tumor agent must show low toxicity on non-neoplastic cells, selectivity index (SI) values were calculated for the most active compounds (Table 2). From the cytotoxicity and selectivity analysis, based on the lowest IC_50_ and highest SI values, it can be concluded that the most sensitive cell line was MCF-7 for the sesquiterpene-aryl derivatives. Compound **8f** elicited the lowest IC_50_ values on three cancer cell lines (IC_50_ = 6.2, 26.2, and 7.1 μM, respectively), but this compound also showed similar toxicity on control cells (SI values less than 1.0). However, **8f** was less active than daunorubicin in three cancer lines, but compared to 5-FU, **8f** was more cytotoxic on MCF-7 and PC-3 cells. Both **6a** and **8c** elicited the highest SI values on MCF-7 cells, although much less than that presented by daunorubicin. However, **6a** was 100-fold more selective than 5-FU in MCF-7 and 20-fold in PC-3., respectively. Likewise, **6a** and **6e** showed similar behavior on HT-29 cells, both compounds were more selective than the reference drugs. Therefore, **6a** was the most promising compound, showing a good potency and selectivity profile in this preliminary study.

Chemically, a slight structure–activity relationship can be proposed between these compounds. Firstly, as observed in **6a**, substitutions on C-3′ of the phenyl ring with a hydroxyl group and a methyl group on C-5′ could be responsible for an increase in cytotoxicity on cancer cells. However, incorporating a methyl ester group on C-4′ (**6a** vs. **6c**) or substituting C-3′ by a methoxy group (**6a** vs. **6b**) decreased the activity. Secondly, substituting a formyl group at C-4′ increased cytotoxicity in all cell lines, as observed for compounds **6e** and **6f**. Finally, for compounds such as -**8**, the substitution of the phenyl ring by three oxygenated groups appeared to be of great importance for the cytotoxic activity observed for compounds **8d**–**g**, with the substitutions on C-2′, -3′, and -4′ being the most relevant, according to the chemical structure of compound **8f**.

### 2.3. Caspase-Glo 3/7 Assay

To determine the cell death type induced by the most promising aryl-drimane esters, we evaluated their capacity to induce apoptosis through the activation of caspases 3 and 7 on MCF-7 cells. For this purpose, **6a**, **6e**, **6f**, **8c**, **8f**, and **8g** were incubated for 24 h on MCF-7 cells at a concentration of 50 μM, close to the IC_50_ values for cytotoxicity in most compounds. The results shown in Figure 3 indicate that all compounds tested induced caspases-3/7 activity significantly over control levels. Compounds **8f** and **8g** showed high caspase activity, 3.5-fold over the control, compared to other drimane derivatives, which correlates with their poor selectivity index profile. In addition, the most selective compounds on MCF-7, **6a,** and **8c** increased the caspases-3/7 activity by two-fold, with respect to the control, as well as compounds **6e** and **6f**. In summary, these drimane derivatives induced apoptosis in the MCF-7 cell line, showing a potential antitumoral effect.

### 2.4. Topoisomerase I Inhibition

Several studies showed that some antitumor compounds that induce apoptosis in cancer cell lines by activating caspases-3/7 are related to the inhibition of TOP1 [30], and the ability of the selected drimane derivatives to inhibit TOP1 activity was determined. In this experiment, daunorubicin was used as control due to this compound being reported early as a TOPI inhibitor [31]. First, the optimal concentration of these compounds for the DNA relaxation assays was established. As shown in Figure 4a, the selected concentration for **6a** was 50 μM, which was used for the rest of the drimane derivatives. In Figure 4b, all compounds inhibited TOP1-induced DNA relaxation. This effect was more prominent for **6a** compared to the other compounds tested. In Figure 4c, it is shown that daunorubicin modified the TOP1-induced DNA relaxation in a minor way than compounds **6a** and **8c**.

### 2.5. Docking Studies

According to the results of these compounds on the ability to inhibit TOP1, especially by compound **6a**, docking studies were carried out to understand the binding mode on this enzyme and the interaction with DNA. A standard protocol was used to perform induced-fit docking and daunorubicin was considered a control because this compound is a well-known TOP1 inhibitor [31]. Figure 5a–c shows a comparison of the best docking poses of compound **6a** and the reference drug daunorubicin in complex with TOP1 and DNA. The XP Glide Score for **6a** was −7.232, while for daunorubicin, it was −12.253. This difference is significant and shows that **6a** did not outperform the reference compound.

Figure 5a illustrates a superposition of the binding mode of daunorubicin and compound **6a** in the complex. Daunorubicin achieved an intercalating accommodation. The extensive planar system was inserted between the DG11 (3.53 Å of distance), DC112 (3.67 Å of distance), and DA113 (3.56 Å of distance) bases, establishing π-stacking interactions. Figure 5b shows a 2D diagram of the interactions and the environment around daunorubicin no more than 4 Å away. Daunorubicin also established ionic and hydrogen-bonding interactions with Glu356 (2.01 Å of distance) through the ammonium group of the aminoglycoside portion. It also established hydrogen bonds with Tyr426 and the OH group of the aminoglycoside portion (2.01 Å of distance), and finally, a hydrogen bond with Lys374 and the oxygen of the acetate group (1.79 Å of distance). This pattern of interactions highlights the importance of the molecule possessing hydrogen-bond donor and acceptor groups, as well as a basic amino group capable of interacting with Glu356.

On the other hand, compound **6a** inserted its benzene ring between the DC112 and DA113 bases (3.7 Å of distance). Figure 5c shows the 2D interaction diagram, and it can be seen that the hydroxyl group of the benzene established a double hydrogen bond with the DA113 base (1.79 Å of distance) and with the Arg364 residue (2.09 Å of distance). Additionally, the drimane fragment established hydrophobic interactions with the residues of Ile377, Lys374, Glu418, Trp416, Glu356, Ile355, Lys354, Asn352, Ile427, Tyr426, and Lys425.

From the preliminary docking study, we can conclude that while compound **6a** did intercalate between the same base pair as daunorubicin, the number of interactions was considerably lower. The insertion of a more extensive aromatic system and the functionalization of drimane with basic amines and hydroxyl groups is needed to improve the number of interactions. In order to evaluate the permanence of compound **6a** in the complex, we then carried out a comparative molecular dynamics study with daunorubicin.

### 2.6. Molecular Dynamics Simulation

In order to evaluate the stability of the complex between compound **6a** and DNA-Topo I, we carried out a molecular dynamics study for **6a** and for the reference pattern daunorubicin. An amount of 100 ns of simulation time was performed and the metrics for both compounds are presented in Figure 6. In the case of daunorubicin (Figure 6a), the protein-DNA complex was stabilized at 10 ns and remained stable throughout the simulation time. The RMSD for the ligand showed a similar behavior. The fluctuations remained within the range of 1 Å throughout the entire time. In the case of compound **6a** (Figure 6b), the RMSD graph showed strong fluctuations for both the complex and the ligand. A stabilization of the fluctuations can be observed from 20 ns onwards; however, between 60 and 80 ns, they increased. The variations in RMSD for the ligand **6a** were much more drastic than for daunorubicin, and fluctuated on average from 2 Å to 8 Å. The total-contact graphs (Figure 6c) showed that the scale of contacts for daunorubicin was from 0 to 9, with most of the contacts between 3 and 6. For compound **6a**, however, the scale worsened (0 to 4), with most of the contacts below 2. Finally, the solvent accessible surface area (SASA) graphs (Figure 6d,e) showed that daunorubicin had a lower exposure to solvent than compound **6a**. From 30 ns onwards, the degree of solvation in daunorubicin decreased, while in **6a**, the solvation increased from 30 ns onwards. All the above leads us to conclude that compound **6a** did not match daunorubicin in its mode of binding to the active site. Furthermore, it seems highly likely that it could exit the binding site or that the disruptive process could take place in another cavity. Figure 6f,g shows the two poses of daunorubicin (Figure 6f) and compound **6a** (Figure 6g) at the end of the 100 ns simulation. As can be seen, the final arrangement of **6a** did not conserve the aromatic ring towards the base groove of the DNA, as daunorubicin does. This reinforces the idea of inserting larger aromatic groups connected to drimane, as well as a basic amine that allows potential anchoring to glutamates.

### 2.7. Physicochemical Properties

Drug-likeness properties such as the pharmacokinetic (ADME) and pharmacodynamics profiles are important during the process of discovery and development of a drug. These properties guide the optimization of a leading compound to a successful candidate for pre-clinical stages [32]. Based on Lipinski’s rules and using SwissADME web server (http://www.swissadme.ch/index.php, accessed on 15 February 2023), **6a** met the criteria for good permeability and bioavailability (Table 3) [33,34]. In addition, the topological polar surface area (TPSA) and rotatable bonds (NRB) were determined for **6a** (Table 3), where, again, this compound showed TPSA values lower than 140 A2 and ≤10 rotatable bonds; this suggests good solubility, high capacity to penetrate cell membranes, and good intestinal absorption. Furthermore, the bioavailability radar chart for **6a** showed that all parameters used for oral absorption prediction were in the desired range (pink region, Figure 7). This radar indicates that: FLEX (Flexibility), LIPO (Lipophilicity), INSOLU (Solubility), SIZE and POLAR (Polarity), and INSATU (Saturation).

## 3. Experimental Section

### 3.1. Chemistry

#### 3.1.1. General Information

All reagents were purchased from commercial suppliers and used without further purification. Melting points were measured on a SMP3 apparatus (Stuart-Scientific, now Merck KGaA, Darmstadt, Germany) and were uncorrected. Optical rotations were measured with a sodium lamp (λ = 589 nm, D line) on a Perkin Elmer 241 digital polarimeter (Perkin Elmer, Waltham, MA, USA) equipped with 1 dm cells at the temperature indicated in each case. ^1^H-, ^13^C-, ^13^C-DEPT-135, gs 2D HSQC, and gs 2D HMBC NMR spectra were recorded in CDCl_3_ and CD_3_OD solutions, and they were referenced to the residual peaks of CHCl_3_ at δ = 7.26 ppm and δ = 77.00 ppm for ^1^H and ^13^C, respectively, and CD_3_OD at δ = 3.30 ppm and δ = 49.00 ppm for ^1^H and ^13^C, on an Advance Neo 400 Digital NMR spectrometer (Bruker, Rheinstetten, Germany) operating at 400.1 MHz for 1H and 100.6 MHz for ^13^C. ESI/MS experiment was carried out on a UHPLC Eksigent1 coupled with MS detector ABSciex1, Triple Quad 4500 model equipment. Thin layer chromatography (TLC) was performed using Merck GF-254 type 60 silica gel. Column chromatography was carried out using Merck type 9385 silica gel. For analytical TLC, silica gel 60 in a 0.25 mm layer was used, and TLC spots were detected by heating after spraying with 10% H_2_SO_4_ in H_2_O.

#### 3.1.2. Synthesis of Drimenal (**2**)

A solution of drimenol (**1**, 300 mg, 1.35 mmol) in CH_2_Cl_2_ (50 mL) PCC (500 mg, 2.32 mmol) was slowly added and the mixture was stirred at 0 °C for 1 h. The mixture was filtrated and evaporated. The residue was purified by silica gel chromatography using hexane/ethyl acetate with increasing polarity to give **2** (yield 78%). ^1^H NMR (400 MHz, CDCl_3_, δ, ppm): 0.85 (3H, s, CH_3_-14), 0.90 (3H, s, CH_3_-13), 1.04 (3H, s, CH_3_-15), 1.65 (3H, bs, CH_3_-12), 2.56 (1H, bs, H-9), 5.67 (1H, bs, H-7), 9.66 (1H, d, *J* = 5,16 Hz, CHO). ^13^C NMR (100 MHz, CDCl_3_, δ, ppm): 15.61 (C-15), 18.18 (C-2), 21.52 (C-14), 21.98 (C-12), 23.55 (C-6), 32.93 (C-4), 33.20 (C-13), 36.91 (C-10), 40.26 (C-1), 41.90 (C-3), 48.95 (C-5), 67.48 (C-9), 125.38 (C-7), 127.68 (C-8), 206.61 (C-11). The NMR spectra signals agreed with the reported values [35].

#### 3.1.3. Synthesis of Drimenic Acid (**3**)

A solution of drimenal (**3**, 500 mg 0, 2.28 mmol.) in *t*-BuOH (50 mL) and 2-methyl-2-butene (11.0 mL) was stirred, treated dropwise with a solution of NaH_2_PO_4_ × 2H_2_O (2.50 g, 16.1 mmol) and NaClO_2_. (2.46 g, 27.2 mmol) in H_2_O (25 mL), stirred at room temperature for 3 h, and evaporated at reduced pressure. The solid was worked up with H_2_O (50 mL) and extracted with CH_2_Cl_2_ (100 mL). The organic phase was concentrated under a vacuum, and the product of the reaction was purified through chromatography using hexane/ethyl acetate with increasing polarity to give **3** (yield 85%). ^1^H NMR (400 MHz, CDCl_3_, δ, ppm): 0.89 (3H, s, CH_3_-14), 0.92 (3H, s, CH_3_-13), 1.00 (3H, s, CH_3_-15), 1.68 (3H, s, CH_3_-12), 2.92 (1H, bs, H-9), 5.57 (1H, bs, H-7). ^13^C (100 MHz, CDCl_3_, δ, ppm): 14.80 (C-15), 18.64 (C-2), 21.29 (C-14), 21.93 (C-12), 23.59, (C-6), 33.03 (C-13), 33.32 (C-4), 36.17 (C-10), 40.28 (C-1), 42.06 (C-3), 49.34 (C-5), 62.01 (C-9), 124.59 (C-7), 128.66 (C-8), 179.29 (C-11). The NMR spectra signals agreed with the reported values [23].

#### 3.1.4. General Procedure to Obtain **6a**–**f**

To a solution of drimenic acid (**3**, 500 mg, 2.12 mmol) in CH_2_Cl_2_ (50 mL), SOCl_2_ (0.3 mL, 4.2 mmol) and pyridine (0.5 mL) were added, the mixture was then stirred at room temperature for 1 h. The corresponding phenol **5a**–**f** was in situ and was then added, and the mixture was stirred at reflux for 24 h. The solvent was concentrated under a vacuum, and the product of the reaction was purified through chromatography using hexane/ethyl acetate with increasing polarity.

(**6a**) 3-Hydroxy-5-methylphenyl-(8a*S*)-2,5,5,8a-tetramethyl-3,4,4a,5,6,7,8,8a-octahydronaphthalene-1-carboxylate; colorless oil, yield 25%, [α]_D_ = +32.1° (c 0.014, CHCl_3_). ^1^H NMR (400 MHz, CDCl_3_, δ, ppm): 0.88 (3H, s, CH_3_-15), 0.91 (3H, s, CH_3_-13), 1.27 (3H, s, CH_3_-14), 1.77 (3H, s, CH_3_-12), 2.15 (2H, m, H-7), 2.30 (3H, s, CH_3_-7′), 6,43 (1H, s, H-2′), 6.50 (2H, s, H-4′ + H-6′). ^13^C NMR (100 MHz, CDCl_3_, δ, ppm): 18.47 (C-2), 18,77 (C-6), 20.59 (C-14), 20.91 (C-12), 21.38 (C-7′), 21.46 (C-15), 32.28 (C-7), 33.15 (C-13), 33.18 (C-4), 36.91 (C-1), 37.15 (C-10), 41.64 (C-3), 50.41 (C-5), 106.39 (C-2′), 113.75 (C-6′), 114.68 (C-4′), 133.70 (C-9), 137.81 (C-8), 140.65 (C-5′), 151.31 (C-1′), 156.19 (C-3′), 168.85 (C-11). HRMS (ESI) calculated for C_22_H_30_O_3_ [M + NH_4_]^+^, 360.2533. Found: 360.2533.

(**6b**) 3-Methoxy-5-methylphenyl-(8a*S*)-2,5,5,8a-tetramethyl-3,4,4a,5,6,7,8,8a-octahydronaphthalene-1-carboxylatewhite; yellow oil, yield 71%, [α]_D_ = +70.0° (c 0.005, CHCl_3_). ^1^H NMR spectrum (400 MHz, CDCl_3_, δ, ppm): 0.88 (3H, s, CH_3_-15), 0.91 (3H, s, CH_3_-13), 1.28 (3H, s, CH_3_-14), 1.78 (3H, s, CH_3_-12), 2.15 (2H, m, H-7), 2.34 (3H, s, CH_3_-7′), 3.79 (3H, s, OCH_3_), 6,47 (1H, s, H-2′), 6.54 (1H, s, H-6′), 6,60 (1H, s H-4′). ^13^C spectrum (100 MHz, CDCl_3_, δ, ppm): 18.49 (C-2), 18,79 (C-6), 20.59 (C-14), 20.93 (C-12), 21.46 (C-15), 21.60 (C-7′), 32.30 (C-7), 33.16 (C-13), 33.20 (C-4), 36.91 (C-1), 37.16 (C-10), 41.66 (C-3), 50.43 (C-5), 104.96 (C-2′), 112.14 (C-6′), 114.75 (C-4′), 133.57 (C-9), 137.92 (C-8), 140.30 (C-5′), 151.39 (C-1′), 160.27 (C-3′), 168.66 (C-11). HRMS (ESI) calculated for C_23_H_32_O_3_ [M + H]^+^, 357.2424. Found: 357.2432.

(**6c**) 3-Hydroxy-4-(methoxycarbonyl)-5-methylphenyl-(8a*S*)-2,5,5,8a-tetramethyl-3,4,4a,5,6,7,8,8a-octahydronaphthalene-1-carboxylate; White solid, yield 43%, m.p. 124.0 °C, [α]_D_ = +27.3° (c 0.022, CHCl_3_). ^1^H NMR spectrum (400 MHz, CDCl_3_, δ, ppm): 0.87 (3H, s, CH_3_-15), 0.91 (3H, s, CH_3_-13), 1.26 (3H, s, CH_3_-14), 1.77 (3H, s, CH_3_-12), 2.15 (2H, m, H-7), 2.54 (3H, s, CH_3_-7′), 3.95 (3H, s, CO_2_CH_3_), 6.51 (1H, dd, *J* = 2.3, 0.6 Hz, H-6′), 6.64 (1H, d, *J* = 2.3 Hz, H-2′), 11.54 (1H, s, OH). ^13^C spectrum (100 MHz, CDCl_3_, δ, ppm): 18.43 (C-2), 18,74 (C-6), 20.58 (C-14), 20.95 (C-12), 21.44 (C-15), 24.19 (C-7′), 32.31 (C-7), 33.13 (C-13), 33.18 (C-4), 36.90 (C-1), 37.18 (C-10), 41.62 (C-3), 50.41 (C-5), 52.18 (CO_2_CH_3_), 108.64 (C-2′), 110.00 (C-4′), 116.66 (C-6′), 134.28 (C-9), 137.54 (C-8), 143.19 (C-5′), 154,78 (C-1′), 164.27 (C-3′), 167.80 (C-11), 171.78 CO_2_CH_3_). HRMS (ESI) calculated for C_24_H_32_O_5_ [M + NH_4_]^+^, 418.2588. Found: 418.2588.

(**6d**) 3-Hydroxy-4-(methoxycarbonyl)-2,5-dimethylphenyl-(8a*S*)-2,5,5,8a-tetramethyl-3,4,4a,5,6,7,8,8a-octahydronaphthalene-1-carboxylate; yellow solid, yield 25%, m.p. 128.0 °C, [α]_D_ = +26.2° (c 0.021, CHCl_3_). ^1^H NMR spectrum (400 MHz, CDCl_3_, δ, ppm): 0.88 (3H, s, CH_3_-15), 0.91 (3H, s, CH_3_-13), 1.29 (3H, s, CH_3_-14), 1.82 (3H, s, CH_3_-12), 2.16 (2H, m, H-7), 2.10 (3H, s, H-8′), 2.50 (3H, s, CH_3_-7′), 3.93 (3H, s, CO_2_CH_3_), 6.47 (1H, s, H-6′), 11.91 (1H, s, OH). ^13^C spectrum (100 MHz, CDCl_3_, δ, ppm): 9.28 (C-8′), 18.35 (C-2), 18,68 (C-6), 20.54 (C-14), 21.04 (C-12), 21.35 (C-15), 23.99 (C-7′), 32.42 (C-7), 33.05 (C-13), 33.07 (C-4), 37.06 (C-1), 37.10 (C-10), 41.51 (C-3), 50.35 (C-5), 51.96 (CO_2_CH_3_), 109.22 (C-4′), 116.26 (C-6′), 116.60 (C-2′), 134.41 (C-9), 137.57 (C-8), 139.07 (C-5′), 153.14 (C-1′), 162.73 (C-3′), 167.73 (C-11), 172.19 CO_2_CH_3_). HRMS (ESI) calculated for C_25_H_34_O_5_ [M + NH_4_]^+^, 432.2745. Found: 432.2744.

(**6e**) 4-Formyl-3-methoxyphenyl-(8a*S*)-2,5,5,8a-tetramethyl-3,4,4a,5,6,7,8,8a-octahydronaphthalene-1-carboxylate; colorless oil, yield 55%, [α]_D_ = +45.0° (c 0.010, CHCl_3_). ^1^H NMR spectrum (400 MHz, CDCl_3_, δ, ppm): 0.88 (3H, s, CH_3_-15), 0.92 (3H, s, CH_3_-13), 1.28 (3H, s, CH_3_-14), 1.79 (3H, s, CH_3_-12), 2.17 (2H, m, H-7), 3.93 (3H, s, OCH_3_), 6.77 (1H, *J* = 1.9 Hz, H-2′), 6.82 (1H, m, H-6′), 7.87 (1H, *J* = 7.9 Hz, H-5′), 10.39 (1H, s, CHO). ^13^C spectrum (100 MHz, CDCl_3_, δ, ppm): 18.41 (C-2), 18,75 (C-6), 20.58 (C-14), 21.07 (C-12), 21.44 (C-15), 32.38 (C-7), 33.14 (C-13), 33.20 (C-4), 36.94 (C-1), 37.25 (C-10), 41.59 (C-3), 50.45 (C-5), 55.88 (O-CH_3_), 105.45 (C-2′), 114.24 (C-6′), 122.54 (C-4′), 129.85 (C-5′), 134.85 (C-9), 137.43 (C-8), 156.80 (C-1′), 162.85 (C-3′), 167.74 (C-11), 188.64 (CHO). HRMS (ESI) calculated for C_23_H_30_O_4_ [M + H]^+^, 371.2217. Found: 371.2217.

(**6f**) 4-Formyl-2-methoxyphenyl-(8a*S*)-2,5,5,8a-tetramethyl-3,4,4a,5,6,7,8,8a-octahydronaphthalene-1-carboxylate; White solid, 236 mg, yield 30%, m.p. 88.5 °C, [α]_D_ = +35.4° (c 0.006, CHCl_3_). ^1^H NMR (400 MHz, CDCl_3_, δ, ppm): 0.88 (3H, s, CH_3_-15), 0.91 (3H, s, CH_3_-13), 1.28 (3H, s, CH_3_-14), 1.85 (3H, s, CH_3_-12), 2.17 (2H, m, H-7), 3.89 (3H, s, OCH_3_), 7.22 (1H, d, *J* = 7.9 Hz, H-6′), 7.48 (1H, dd, *J* = 7.9, 1.7 Hz, H-5′), 7.50 (1H, d, *J* = 1.5 Hz, H-3′), 9.94 (1H, s, CHO). ^13^C NMR (100 Mz, CDCl_3_, δ, ppm): 18.44 (C-2), 18.81 (C-6), 20.49 (C-14), 20.87 (C-12), 21.44 (C-15), 32.38 (C-7), 32.46 (C-4), 33.15 (C-13), 36.81 (C-1), 37.13 (C-10), 41.61 (C-3), 50.42 (C-5), 55.81 (O-CH_3_), 110.82 (C-3′), 123.60 (C-6′), 124.67 (C-5′), 134.80 (C-9), 135.11 (C-4′), 137.27 (C-8), 144.87 (C-1′), 152.27 (C-2′), 167.28 (C-11), 191.04 (CHO). HRMS (ESI) calculated for C_23_H_30_O_4_ [M + H]^+^, 371.2217. Found: 371.2222.

#### 3.1.5. General Procedure to Obtain **8a**–**g**

Benzoic acid chlorides **7a**–**g** were obtained by the addition of SOCl_2_ (0.3 mL, 4.2 mmol) and pyridine (0.5 mL) to the corresponding commercially available benzoic acid (2.5 mmol) dissolved in CH_2_Cl_2_ (50 mL). The mixture was stirred at room temperature for 1 h before drimenol (1, 556 mg, 2.5 mmol) was added in situ and the mixture was stirred at reflux for 24 h. The solvent was concentrated under a vacuum, and the product of the reaction was purified through chromatography using hexane/ethyl acetate with increasing polarity.

(**8a**) ((1*S*,8a*S*)-2,5,5,8a-tetramethyl-1,4,4a,5,6,7,8,8a-octahydronaphthalen-1-yl)methyl-3,5-dimethoxybenzoate; yellow oil, 908 mg, yield 94%, [α]_D_ = +26.6° (c 0.021, CHCl_3_). ^1^H NMR (400 MHz, CDCl_3_, δ, ppm): 0.88 (3H, s, CH_3_-13), 0.90 (6H, s, CH_3_-14 and CH_3_-15), 1.74 (3H, d, *J* = 1.0 Hz, CH_3_-12), 2.03 (2H, m, H-6), 2.17 (1H, m, H-9), 3.82 (6H, s, 2 × OCH_3_), 4.29 (1H, dd, *J* = 11.6, 6.1 Hz, H-11), 4.55 (1H, dd, *J* = 11.6, 3.4 Hz, H-11), 5.54 (1H, bs, H-7), 6.64 (1H, t, *J* = 2.4 Hz, H-4′), 7.17 (2H, d, *J* = 2.4 Hz, H-2′ and H-6′). ^13^C NMR (100 MHz, CDCl_3_, δ, ppm): 14.70 (C-15), 18.72 (C-2), 21.89 (C-14), 21.95 (C-12), 23.62 (C-6), 32.96 (C-4), 33.31 (C-13), 36.02 (C-10), 39.65 (C-1), 42.05 (C-3), 49.86 (C-5), 53.57 (C-9), 55.51 (2 × OCH_3_), 63.67 (C-11), 105.48 (C-4′), 107.12 (C-2′ and C-6′), 123.88 (C-7), 132.32 (C-1′), 132.35 (C-8), 160.60 (C-3′ and C-5′), 166.30 (CO_2_). HRMS (ESI) calculated for C_24_H_34_O_4_ [M + H]^+^, 387.2530. Found: 387.2528.

(**8b**) ((1*S*,8a*S*)-2,5,5,8a-tetramethyl-1,4,4a,5,6,7,8,8a-octahydronaphthalen-1-yl)methyl 3,5-dimethoxy-4-methylbenzoate; yellow oil, 338 mg, yield 35%, [α]_D_ = +32.1° (c 0.014, CHCl_3_). ^1^H NMR (400 MHz, CDCl_3_, δ, ppm): 0.87 (3H, s, CH_3_-13), 0.90 (3H, s, CH_3_-15), 0.91 (3H, s, CH_3_-14), 1.75 (3H, d, *J* = 0.8 Hz, CH_3_-12), 2.12 (3H, s, 7′), 2.24 (1H, m, H-9), 3.85 (6H, s, 2 × OCH_3_), 4.28 (1H, dd, *J* = 11.6, 6.1 Hz, H-11), 4.56 (1H, dd, *J* = 11.6, 3.4 Hz, H-11), 5,55 (1H, bs, H-7), 7.22 (2H, s, H-2′ and H-6′). ^13^C NMR (100 MHz, CDCl_3_, δ, ppm): 8.60 (C-7′), 14.63 (C-15), 18.73 (C-2), 21.91 (C-14), 21.94 (C-12), 23.62 (C-6), 32.95 (C-4), 33.30 (C-13), 36.01 (C-10), 39.67 (C-1), 42.04 (C-3), 49.85 (C-5), 53.59 (C-9), 55.72 (2 × OCH_3_), 63.47 (C-11), 104.55 (C-2′ + C-6′), 120.20 (C-4′), 123.86 (C-7), 128.52 (C-1′), 132.43 (C-8), 158.00 (C-3′ and C-5′), 166.63 (CO_2_). HRMS (ESI) calculated for C_25_H_36_O_4_ [M + H]^+^, 401.2686. Found: 401.2686.

(**8c**) ((1*S*,8a*S*)-2,5,5,8a-tetramethyl-1,4,4a,5,6,7,8,8a-octahydronaphthalen-1-yl)methyl-3,5-dihydroxy-4-methylbenzoate; white solid, 335 mg, yield 36%, m.p. 98.5 °C, [α]_D_ = +52.6° (c 0.004, CHCl_3_). ^1^H NMR (400 MHz, CD_3_OD, δ, ppm): 0.88 (3H, s, CH_3_-13), 0.93 (6H, s, CH_3_-14 and CH_3_-15), 1.74 (3H, s, CH_3_-12), 2.06 (3H, s, 7′), 2.21 (1H, m, H-9), 4.28 (1H, dd, *J* = 11.8, 5.5 Hz, H-11), 4.46 (1H, dd, *J* = 11.8, 3.5 Hz, H-11), 5.54 (1H, bs, H-7), 6.95 (2H, s, H-2′ and H-6′). ^13^C NMR (100 MHz, CD_3_OD, δ, ppm): 9.03 (C-7′), 15.47 (C-15), 19.93 (C-2), 22.31 (C-14), 22.53 (C-12), 24.82 (C-6), 34.00 (C-13), 34.05 (C-4), 37.37 (C-10), 41.06 (C-1), 43.40 (C-3), 48.5 (C-5), 53.59 (C-9), 64.24 (C-11), 108.49 (C-2′ and C-6′), 118.49 (C-4′), 124.95 (C-7), 129.46 (C-1′), 133.78 (C-8), 157.66 (C-3′ and C-5′), 168.64 (CO_2_). HRMS (ESI) calculated for C_23_H_32_O_4_ [M + H]^+^, 373.2373. Found: 373.2375.

(**8d**) ((1*S*,8a*S*)-2,5,5,8a-tetramethyl-1,4,4a,5,6,7,8,8a-octahydronaphthalen-1-yl)methyl-4-hydroxy-3,5-dimethoxybenzoate; white solid, 773 mg, yield 80%, m.p. 105.0 °C, [α]_D_ = +28.7° (c 0.023, CHCl_3_). ^1^H NMR spectrum (400 MHz, CDCl_3_, δ, ppm): 0.88 (3H, s, CH_3_-13), 0.91 (6H, s, CH_3_-14 and CH_3_-15), 1.75 (3H, d, *J* = 1.0 Hz, CH_3_-12), 2.18 (1H, m, H-9), 3.92 (6H, s, 2 × OCH_3_), 4.27 (1H, dd, *J* = 11.6, 6.2 Hz, H-11), 4.56 (1H, dd, *J* = 11.6, 3.4 Hz, H-11), 5,55 (1H, bs, H-7), 5.89 (1H, s, OH), 7.31 (2H, s, H-2′ + H-6′). ^13^C spectrum (100 MHz, CDCl_3_, δ, ppm): 14.65 (C-15), 18.74 (C-2), 21.93 (C-14), 21.95 (C-12), 23.64 (C-6), 33.0 (C-4), 33.30 (C-13), 36.04 (C-10), 39.68 (C-1), 42.06 (C-3), 49.87 (C-5), 53.63 (C-9), 56.33 (2 × OCH_3_), 63.49 (C-11), 106.54 (C-2′ + C-6′), 121.47 (C-1′), 123.90 (C-7), 132.43 (C-8), 139.10 (C-4′), 146.60 (C-3′ + C-5′), 166.30 (CO_2_). HRMS (ESI) calculated for C_24_H_34_O_5_ [M + H]^+^, 402.2241. Found: 403.2480.

(**8e**) ((1*S*,8a*S*)-2,5,5,8a-tetramethyl-1,4,4a,5,6,7,8,8a-octahydronaphthalen-1-yl)methyl-3,4,5-trimethoxybenzoate; colorless oil, 676 mg, yield 70%, [α]_D_ = +27.5° (c 0.005, CHCl_3_). ^1^H NMR spectrum (400 MHz, CDCl_3_, δ, ppm): 0.87 (3H, s, CH_3_-13), 0.91 (6H, s, CH_3_-14 and CH_3_-15), 1.74 (3H, d, *J* = 0.8 Hz, CH_3_-12), 2.18 (1H, m, H-9), 3.89 (6H, s, 2 × OCH_3_), 3.90 (3H, s, OCH_3_), 4.28 (1H, dd, *J* = 11.6, 6.1 Hz, H-11), 4.55 (1H, dd, *J* = 11.6, 3.4 Hz, H-11), 5.55 (1H, bs, H-7), 7.28 (2H, s, H-2′ + H-6′). ^13^C spectrum (100 MHz, CDCl_3_, δ, ppm): 14.67 (C-15), 18.73 (C-2), 21.91 (C-14), 21.95 (C-12), 23.62 (C-6), 32.96 (C-4), 33.30 (C-13), 36.03 (C-10), 39.68 (C-1), 42.03 (C-3), 49.85 (C-5), 53.59 (C-9), 56.12 (2 × OCH_3_), 60.89 (OCH_3_), 63.61 (C-11), 106.67 (C-2′ + C-6′), 123.95 (C-7), 125.48 (C-1′), 132.33 (C-8), 142.05 (C-4′), 152.89 (C-3′ + C-5′), 166.13 (CO_2_). HRMS (ESI) calculated for C_25_H_36_O_5_ [M + H]^+^, 417.2636. Found: 417.2638.

(**8f**) ((1*S*,8a*S*)-2,5,5,8a-tetramethyl-1,4,4a,5,6,7,8,8a-octahydronaphthalen-1-yl)methyl-2,3,4-trimethoxybenzoate; yellow oil, 502 mg, yield 52%, [α]_D_ = +23.0° (c 0.007, CHCl_3_). ^1^H NMR spectrum (400 MHz, CDCl_3_, δ, ppm): 0.86 (3H, s, CH_3_-13), 0.87 (3H, s, CH_3_-15), 0.89 (3H, s, CH_3_-14), 1.74 (3H, d, *J* = 0.8 Hz, CH_3_-12), 2.15 (1H, m, H-9), 3.86 (3H, s, 3′-OCH_3_), 3.89 (3H, s, 4′-OCH_3_), 3.93 (3H, s, 2′-OCH_3_), 4.30 (1H, dd, *J* = 11.7, 5.9 Hz, H-11), 4.47 (1H, dd, *J* = 11.7, 3.2 Hz, H-11), 5.52 (1H, bs, H-7), 6.68 (1H, d, *J* = 8.9 Hz, H-3′), 7.57 (1H, d, *J* = 8.9 Hz, H-2′). ^13^C spectrum (100 MHz, CDCl_3_, δ, ppm): 14.65 (C-15), 18.69 (C-2), 21.80 (C-14), 21.92 (C-12), 23.59 (C-6), 32.93 (C-4), 33.28 (C-13), 36.00 (C-10), 39.59 (C-1), 42.06 (C-3), 49.85 (C-5), 53.56 (C-9), 56.01 (4′-OCH_3_), 60.95 (3′-OCH_3_), 61.75 (2′-OCH_3_), 63.20 (C-11), 106.76 (C-5′), 118.07 (C-1′), 123.60 (C-7), 126.78 (C-6′), 132.64 (C-8), 142.90 (C-3′), 154.85 (C-2′), 157.01 (C-4′), 165.33 (CO_2_). HRMS (ESI) calculated for C_25_H_36_O_5_ [M + H]^+^, 417.2636. Found: 417.2625.

(**8g**) ((1*S*,8a*S*)-2,5,5,8a-tetramethyl-1,4,4a,5,6,7,8,8a-octahydronaphthalen-1-yl)methyl-2,4,5-trimethoxybenzoate; yellow oil, 338 mg, yield 35%, [α]_D_ = +19.9° (c 0.020, CHCl_3_). ^1^H NMR (400 MHz, CDCl_3_, δ, ppm): 0.85 (3H, s, CH_3_-13), 0.87 (6H, s, CH_3_-15 and CH_3_-14), 1.74 (3H, bs, CH_3_-12), 2.13 (1H, m, H-9), 3.82 (3H, s, 2′-OCH_3_), 3.86 (3H, s, 5′-OCH_3_), 3.91 (3H, s, 4′-OCH_3_), 4.23 (1H, dd, *J* = 11.7, 5.8 Hz, H-11), 4.50 (1H, dd, *J* = 11.7, 3.5 Hz, H-11), 5.51 (1H, d, *J* = 1.8 Hz, H-7), 6.50 (1H, s, H-3′), 7.40 (1H, s, H-6′). ^13^C NMR (100 MHz, CDCl_3_, δ, ppm): 14.31 (C-15), 18.43 (C-2), 21.54 (C-14), 21.64 (C-12), 23.31 (C-6), 32.60 (C-4), 35.71 (C-10), 39.34 (C-1), 41.77 (C-3), 49.55 (C-5), 53.32 (C-9), 55.65 (4′-OCH_3_), 55.86 (2′-OCH_3_), 56.38 (5′-OCH_3_), 62.47 (C-11), 97.17 (C-3′), 110.28 (C-1′), 114.04 (C-6′), 123,22 (C-7), 132.42 (C-8), 142,04 (C-2′), 153,17 (C-4′), 155, 52 (C-5′), 165,03 (CO_2_).HRMS (ESI) calculated for C_25_H_36_O_5_ [M + H]^+^, 417.2636. Found: 417.2638.

### 3.2. Biological Evaluation

#### 3.2.1. In Vitro Growth Inhibition Assay

The sulforhodamine B assay was used. Briefly, the cells were set up at 5 × 10^3^ cells per well of a 96-well microplate for 18 h. Cells were incubated at 37 °C in a humidified 5% CO_2_/95% air mixture and treated with the compounds at different concentrations for 72 h. At the end of drug exposure, cells were fixed with 50% trichloroacetic acid at 4 °C. After washing with distilled water, cells were stained with 0.1% sulforhodamine B, dissolved in 1% acetic acid (50 μL/well) for 30 min, and, subsequently, washed with 1% acetic acid to remove the unbound stain. The protein-bound stain was solubilized with 100 μL of 10 mM unbuffered Tris base, and the cell density was determined using a fluorescence plate reader (wavelength 540 nm). Results are expressed as IC_50_ values (μM) ± SD.

#### 3.2.2. Caspase-Glo 3/7 Assay

The activity of caspase 3/7 in treated MCF-7 cells was determined using a Caspase-Glo-3/7 Assay kit (Promega, UK—Cat. No G8090). The caspase-Glo 3/7 assay is a homogenous, luminescent assay that measures caspase-3 and -7 activities, both key-players in apoptosis. The addition of the Caspase-Glo reagent resulted in cell lysis, followed by caspase cleavage of caspase-3/7 substrate and the generation of a luminescent signal produced by luciferase. Luminescence was proportional to the amount of caspase present. MCF-7 cells (1 × 10^4^ cells per well) were grown in a 96-well white plate and were allowed to adhere overnight. The cells were then treated for 24 h with the synthesized compounds (50 µM), with the inclusion of a blank as a control. After treatment, the Caspase-Glo^®^ 3/7 was prepared according to the manufacturer’s guidelines. Briefly, 100 μL of Caspase-Glo 3/7 reagent was added to 100 μL of culture medium containing the cells previously treated in each well. The content was mixed gently by placing the plate on a shaker for 30 s, and the plate was incubated at room temperature in the dark for 1 h before luminescence was read on a luminometer (Berthold Sirius Single Tube Luminometer, Pforzheim, Germany). Fold change in caspase activity was obtained by first subtracting the luminescence value for the blank well from every other value, and then, expressing the average luminescence of each treatment as a fold change with respect to the average luminescence of the negative control.

#### 3.2.3. Assay of Topoisomerase I Activity

Topoisomerase I activity was assayed by the relaxation of supercoiled DNA (pEGFP-N1) as the manufacturer’s instructions indicated. Human recombinant Topoisomerase I and buffer were from (Sigma-Aldrich). An amount of 2 μL of 10× topoisomerase I reaction buffer and 200 ng plasmid DNA and the indicated concentrations of compounds were incubated 30 min at 37 °C in Topo I buffer (500 mM Tris·HCl, pH 7.5, 1 M KCl, 10 mM dithiothreitol, 100 mM EDTA, 50 μg/mL acetylated bovine serum albumin (BSA)) in a total volume of 20 μL. The reaction was stopped by the addition of 5 μL of 5× loading dye (composition). Samples were electrophoresed on an 0.8% agarose gel without ethidium bromide for 2–3 h at 5 to 10 V/cm. The gel was stained with ethidium bromide and photographed with a UV transilluminator.

### 3.3. Molecular Docking and Molecular Dynamics Simulations

#### 3.3.1. Induced Fit Docking

The docking studies were carried out in the Induced Fit Docking module incorporated in Schrödinger suite. The Topoisomerase I, DNA, and topotecan complex (PDB ID = 1K4T) were downloaded using the Protein preparation wizard module. The protein was pre-processed at pH 7.0 +/− 2.0. All water molecules, ions, and other compounds were removed from the crystallized structure. Subsequently, the internal hydrogen bonds of the complex were refined with the “optimize” option, and the final minimization was carried out with the OPLS3e force field.

On the other hand, the compounds were minimized with the LigPrep module at pH = 7.0 +/− 2.0. The chirality of each stereocenter was retained, and compounds were minimized with the OPLS3e force field. Finally, the chirality of each atom was manually determined to avoid inversions in the configuration. The final docking of the composites was executed in the Induced Fit Docking module. The topotecan site was used as the grid center, and the box was defined as 20 Å cubic. The analysis was run with the OPLS3e force field and with an XP precision level. The results were processed manually in the visualization module of the suite.

#### 3.3.2. Molecular Dynamics

Simulation studies were carried out in the Desmond module. For this effect, the best docking pose of the complex with compound **6a** was used. The simulation was set to 100 ns, 1000 frames, 1 atm of pressure, and 300 K using the NPT ensemble. The last simulation pose was manually extracted and analyzed with the suite’s visualizer.

## 4. Conclusions

Two new series of sesquiterpene-aryl derivatives **6a**–**f** and **8a**–**g** were synthesized from (-) drimenol. Cytotoxic activity results showed that compounds **6a**, **6e**, **6f**, **8c**, **8f**, and **8g** inhibited cell growth in MCF-7, HT-29, and PC-3 cancer cells, being more active on the MCF-7 line with IC_50_ values between 6.2 and 38.3 μM. Compound **6a** especially stood out due to its higher selectivity than the reference drug 5-FU in MCF-7 and PC-3. In addition, there was evidence that these six compounds have the potential to inhibit TOP1 and induce apoptosis by activation of caspase-3/7. We found that **6a** was the most promising compound for its cytotoxic effect (IC_50_ 33.7 μM) and selectivity index (SI = 2.0) on MCF-7 cancer cell line. The induced docking study of **6a** showed that it formed a complex with DNA-TOP1 with a binding mode similar to topotecan. The aromatic ring interacted via pi-stacking interaction with DNA base 113, while the drimane system interacted with the protein through multiple hydrophobic interactions. A molecular dynamics study showed that **6a** generated water-mediated hydrogen bonding interactions with Arg364, Asn722, and Lys751. However, its interaction profile was inferior to daunorubicin over the simulation time. We suggest exploring the insertion of a naphthalene system connected to the drimane. Finally, predictions regarding ADME properties would indicate that **6a** has a potential good oral bioavailability; therefore, considering these results, drimane scaffold should be considered in the search for new antitumor agents.

## Data Availability

The data presented in this study are available in Appendix A.

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
