# Peer review of "Cytotoxic Activity, Topoisomerase I Inhibition and In Silico Studies of New Sesquiterpene-aryl Ester Derivatives of (-) Drimenol"

_molecules, 2023, doi:10.3390/molecules28093959_

Round 1

Reviewer 1 Report

-          It will be valuable to add in the reasons behind the chemical modifications made.

-          Are the synthesised compounds more active than the parent compounds? This needs to be shown that they are indeed more active before further consideration for publication.  

-          Well-known top I inhibitor required as control in top I assay.

Author Response

Thank you for reconsidering our submission and for sending your comments. We have considered all the observations and have taken appropriate action to address the corrections of our manuscript. Attached are our responses to your comments and a description of the changes we have made to the manuscript. We believe that in this revised version we have responded to the proposed comments. Below you will find our point-by-point responses, respectively. Our responses and the text included in the new version of the manuscript are highlighted in yellow. The revised version of the manuscript and supporting information have been edited. Once again, we reiterate our gratitude for the comments on our study.

Yours sincerely,

Dr. Mauricio Cuellar

Reviewer 2 Report

The manuscript “Cytotoxic activity, topoisomerase I inhibition and in silico studies of new sesquiterpene-aryl ester derivatives of (-) drimenol” report the synthesis sesquiterpene-aryl derivatives and their antitumoral proprieties in vitro and in silico. The structures obtained not presented very high biological activity, but the experiments are well designed. I suggest this paper could be reconsidered for publication after major revision:

1)     Section 1 (line 42) – indicate the family name of the genus. It must be take into account in all the text.

2)     Section 2.2 – which compound has been used as control?  Please, include this information in Table 1.

3)     Section 2.3 – which is the control described in Figure 3? Please, include this information in the subtitles.

4)    Section 5 (line 539) – Attention if the word “significantly”. It has been applied some statistical methodology in the results?

Author Response

(The authors gave the same response as above.)

Reviewer 3 Report

The authors evaluated two sets of drimenol derivatives in this study and proposed that the most promising derivative, compound 6a, binds to TOP1-DNA to achieve its anti-tumor activity.

One obvious issue for these compounds is that their best IC50 values are close to their solubility limit, as the authors mentioned “depending on solubility” when testing compounds for inhibition on tumor cell growth. The authors also tested compounds at up to 50 uM for caspases 3 and 7 activation assay and TOP1 inhibition assay. It is entirely possible that these compounds already form aggregates at the testing concentration and the inhibitory effects from these compounds are due to non-specific binding to possibly multiple sites on a single or multiple targets (including TOP1-DNA). The authors should discuss such possibilities in their manuscript.

For TOP-1 inhibition assay, can the authors explain why not also include topotecan as a positive control? This would help their comparison in docking studies, so that we can better gauge the accuracy and relevance of docked conformations. At the very least, the authors should compare some experimental data of 6a vs topotecan.

It would be much better if the authors can perform the same MD simulations for their control inhibitor, topotecan, and compare the results with 6a. This is especially useful given compound 6a seems to be losing interactions with the target after MD simulations. The authors should also include analyses of solvent accessible surface area of the inhibitor over the MD trajectory, possibly in the supplementary.

Subsection 2.7 can go into the supplementary. I think the most valuable indicator here is cLogP, which confirmed the poor water solubility of 6a.

I also suggest the authors show atom numbering for at least one compound of each set.

Author Response

(The authors gave the same response as above.)

Round 2

Reviewer 1 Report

The authors have addressed the comments and the manuscript is recommended for publication.

Author Response

Dear Reviewer
Thank you very much for the comments and input to our manuscript.

Kind regards,

Mauricio Cuellar

Reviewer 2 Report

The authors carried out the corrections suggested. I consider that the manuscript is able to be published in the present form. 

Author Response

(The authors gave the same response as above.)

Reviewer 3 Report

The authors made some improvements, with the notable addition of a known topoisomerase daunorubicin as a control, to the original manuscript.

However, it is very difficult to see the comparison of daunorubicin vs 6a in their molecular modeling calculations. Except for the docking score (-12.256 vs -7.232), I do not see a direct head-to-head comparison. Why not include the docked pose of both compounds in Figure 5? Why not plot the total number of contacts (instead of residue-wise contacts) for TOP1-6a vs TOP1-daunorubicin over simulation time? It would be much easier to capture the differences this way and the authors should focus on the comparison between compounds instead of among residues. The scale of the axis “Interactions Fraction” showed 6a is probably worse than daunorubicin anyway. The authors can do the same head-to-head comparison with SASA. Since the authors are not deriving binding energetics from MD trajectories, they could also evaluate the geometric stability of 6a vs daunorubicin binding to the protein using classical metrics such as RMSD (of ligands relative to the protein), especially considering metrics such as number of contacts and SASA are biased depending on the size of the compound.

It appears to me that 6a does not have strong interaction with TOP1 and its much-worse docking score and egress from the binding site during MD simulations is good evidence. Together with its high logP value, the authors should definitely acknowledge the possibility that 6a does not specifically bind to the target, even though 6a is soluble beyond 0.1 mM.

Author Response

Dear reviewer

Please find enclosed our responses to your comments and suggestions.  Below you will find our point-by-point responses, respectively. Our responses and the text included in the new version of the manuscript are highlighted in yellow. The revised version of the manuscript and supporting information have been edited.
We reiterate our thanks
Kind regards,

Mauricio Cuellar
